# Fast and Noninvasive Hair Test for Preliminary Diagnosis of Mood Disorders

**DOI:** 10.3390/molecules27165318

**Published:** 2022-08-20

**Authors:** Magdalena Świądro-Piętoń, Kai A. Morawiec, Anna Wójtowicz, Sara Świądro, Rafał Kurczab, Dominika Dudek, Renata Wietecha-Posłuszny

**Affiliations:** 1Laboratory for Forensic Chemistry, Department of Analytical Chemistry, Faculty of Chemistry, Jagiellonian University, 2, Gronostajowa St., 30-387 Kraków, Poland; 2Faculty of Mathematical and Natural Science, Department of Chemistry, University of Applied Sciences in Tarnow, 8 Mickiewicza St, 33-100 Tarnów, Poland; 3Department of Adult Psychiatry, Jagiellonian University Medical College, 21a, Mikołaja Kopernika St., 31-501 Kraków, Poland

**Keywords:** hair analysis, drug analysis, ATR-FTIR spectroscopy, depression, PCA

## Abstract

The main objective of this study was to develop a test for the fast and noninvasive prediagnosis of mood disorders based on the noninvasive analysis of hair samples. The database included 75 control subjects (who were not diagnosed with depression) and 40 patients diagnosed with mood disorders such as depression or bipolar disorder. Both women and men, aged 18–65 years, participated in the research. After taking the hair samples, they were washed (methanol–water–methanol by shaking in a centrifuge for two min) and air-dried in a fume hood. Each hair collection was analyzed using Fourier transform infrared spectroscopy attenuated total reflection (ATR-FTIR) spectroscopy. Subsequently, the results obtained were analyzed based on chemometric methods: hierarchical cluster analysis (HCA) and principal component analysis (PCA). As a results of the research conducted, potential differences were noticed. There was a visible change in the spectra intensity at around 2800–3100 cm^−1^ and smaller differences around 1460 cm^−1^; the bands can be assigned to protein vibrations. However, these are preliminary studies that provide a good basis for the development of a test for the initial diagnosis of mood disorders.

## 1. Introduction

Mood disorders include bipolar affective disorder, which is characterized by a depressive episode (decreased activity, pessimism, and sadness), or/and a manic episode (grandiosity and elevated mood). Unipolar affective disorder, a form of depression, is another mood disorder. Each episode seriously disrupts and adversely affects the person’s life. Currently, they are the most common mental health disorders in the world [1,2]. In 2019, depression was one of the most common and chronic diseases in the Polish population health surveys. According to data from the Central Statistical Office, more than 16% of the respondents experienced symptoms that may indicate the presence of depression at various levels of severity [3]. In all age groups studied, women most frequently declared the occurrence of symptoms indicative of depression. Women are much more likely (in a ratio of 2.5:1) to suffer from depression than men. The research also indicated that the percentage of people who declared illnesses that can be a sign of the disease increased with the age of the respondents. Absence or inadequate treatment related to problems with proper diagnosis can increase the risk of relapse disorders and even lead to death [4]. Mood disorders cannot be underestimated, and therapy should be started as soon as possible. Unfortunately, the initial diagnosis of mood disorder is sometimes quite problematic due to the presence of a depressive episode in both unipolar and bipolar disorders. Currently, diagnosis is made only on the basis of an interview and analysis of the patient’s clinical picture, which contributes to a biased diagnosis of the disease [4]. Therefore, it is important to search for quick methodologies using samples such as hair.

Hair is a biological matrix mainly consisting of proteins (65–95%), in particular keratin, as well as glycine, threonine, and many others. The internal structure of hair is complex and contains a loosely packed medulla, a disordered region near the center of the hair surrounded by the cortex, which contains the major part of the fiber mass, mainly consisting of keratin proteins and structural lipids. Hair is a reliable long-term biological indicator tissue used to assess the health of body bioelements and metabolism [5,6]. Moreover, hair fibers are the most desirable biological matrix because they have a wide detection window, which enables the provision of information from several months or even years ago, while blood and urine detection only provides information for a few days. Due to this, hair analysis is used in drug detection and in screening for medication intake by criminals or other people [7].

The studies carried out so far have shown that people with a severe form of depression (after a suicide attempt) have a different multielement hair profile than healthy people. This may be a consequence of the neuroendocrine system, including the hypothalamic-pituitary-thyroid (HPT) and hypothalamus–pituitary–adrenal (HPA) axes, whose dysregulation has been reported. The research also showed that women are much more likely (in a ratio of 2.5:1) to suffer from depression than men [8,9]. Interestingly, patients with a current diagnosis of comorbid depression and anxiety have moderately higher cortisol levels than patients with depression or anxiety alone, or patients in the remission and control groups, which may indicate a chronic state of hyperactivation of the HPA axis [10].

Research has already been conducted on the use of ATR-FTIR spectroscopy to study hair samples. For example, it has been applied in analysis of samples from patients with breast cancer. Breast cancer patients showed an increase in the band maximum ratio at 1446–1456 cm^−1^ for bending vibrations of the C–H in the ATR-FTIR spectra of a single hair fiber [11]. This confirms the possibility of using the fast and noninvasive ATR technique for hair analysis. Therefore, the aim of this study was to develop a fast and noninvasive methodology based on ATR-FTIR spectroscopy for the diagnosis of mood disorders, which may be applied in clinical practice in the future.

## 2. Results and Discussion

### 2.1. Hair FTIR Spectrum

Hair is mainly composed of scleroproteins called keratin, melanin, and a thin layer of lipids. These components determine the hair ATR-FTIR spectrum. A protein spectrum usually has nine main bands called amide bands; see Table 1.

Five amide bands are easily recognizable. The amide I band, which occurs in the 1690–1600 cm^−1^ region, is mostly an effect of C=O stretching vibrations (70–80%), a small contribution of N-H wagging (10–20%), and C–N stretching. Amide II, located at 1575–1480 cm^−1^, is caused by two vibrations: stretching of C–N bond (20–40%) and N-H wagging (40–60%). Amide III (1301–1229 cm^−1^) has an intensity lower than that of amide I and amide II, and is due to C–H stretching (40%), N-H wagging (30%), and CH_3_–C stretching. On the other side of the spectrum (3300–3100 cm^−1^), there are two bands called amide A and amide B. Amide A is an overtone of the amide II band, and amide B is associated with N-H stretching vibrations. All amide bands contain information about the secondary structure of the proteins, and they are sensitive to changes in this structure. There are two other bands at 3060–2850 and 1470–1400 cm^−1^, which originate from vibrations of the CH_2_/CH_3_ groups. Other absorption bands observed are aliphatic fragments of C–H long-chain saturated and unsaturated fatty acids, alcohols, and esters. A broad- and medium-intensity band between 3430 and 3090 cm^−1^ indicates the presence of carboxylic acid (–COOH). Two sharp medium- to high-intensity bands at about 1106 cm^−1^ and 991 cm^−1^ are characteristic of the O–C stretching vibration of the ester functional groups. The strong and broad band at 1037 cm^−1^ corresponds to the C–O stretching vibration of the alcohol (–OH) groups [12,13,14,15].

There are two hair components that may be responsible for these CH_2_/CH_3_ bands: lipids (in carbon chains) and proteins (inside chains of amino acids). Hair spectra taken from the same person are similar, even if the hair was taken from different parts of the head [13]. It is thought that several factors, such as intense hair treatment, diet, and health of the person, can damage the structure of hair proteins, and the damage can be visible in the hair spectrum. It was confirmed that split hair gives significantly different spectra than normal hair. The age and sex of the sample donor do not influence the hair spectrum [13].

### 2.2. Average Spectra

Most hair samples were measured in triplicate on an ATR-FTIR spectrometer. In a few cases, when the amount of hair was insufficient, only one measurement was taken. Altogether, 376 spectra were collected. After preprocessing (baseline correction and normalization), 35 spectra were discarded that did not resemble the typical protein spectra due to inadequate intensity of the main bands. The 341 remaining spectra corresponded to 115 hair samples: 75 samples from the control group and 40 from the treatment group. To find differences between the groups, the spectra of each group were averaged. The results are presented in Figure 1.

In the case of the average spectra, no changes in the main bands were observed, because the main amide bands were almost the same in both groups. As mentioned above, these bands are particularly sensitive to changes in the secondary structure of proteins. It means that the differences observed in the treatment group were not related to the basic protein structure. There were two regions with differences between the control and treatment groups. In the 3100–2800 cm^−1^ region, the treatment group bands had a lower intensity. In this region, there was mainly the C–H stretching band and some of the amide B band. The second difference occurred in the 1470–1400 cm^−1^ region, which corresponds to the C–H deformation vibrations. The cause of these differences may be the same, because, in both cases, changes in the spectrum concern the same chemical groups: CH_2_ and CH_3_. In proteins, these groups appear only on the side chains of amino acids, and they are not responsible for any major interactions in the protein, such as hydrogen or disulfide bonds. These CH_2_ and CH_3_ bands are also typical of lipid spectra, which suggests a lower concentration of lipids in the treatment group. The averaged spectra in the ranges in which changes were observed are shown in Figure 2.

It is important to note that the standard deviation of the control group spectra was relatively high. Hair is a specific type of biological matrix, and the structure of hair proteins is influenced by the medical history of a person. It is also possible that members of the control group had some episodes of untreated depression or strong emotions that could have had the same effects on the spectrum as those seen in the treatment group. 

In Figure 3, the first (A–C) and second derivatives (D–F) of the mean spectra are presented, obtained for the results shown in Figure 1 and Figure 2.

### 2.3. HCA and PCA

To verify conclusions about the differences and to check if there were any similarities in the spectra of the treatment group spectra, three hierarchical cluster analysis (HCA) dendrograms were prepared [16] from the spectra of the same hair.

The first dendrogram containing two main clusters (Figure 4A) was carried out to check the impact differences in the mean spectra in the 3100–2800 and 1470–1400 cm^−1^ regions. The red dashed lines characterize the treatment group, whereas the blue dashed lines indicate the control group. The dendrogram shows that the data obtained for the control and treatment groups were partially mixed. However, we observed that almost all spectra in the cluster on the right are from the control group. Moreover, the treatment group spectra are close to each other in their cluster. These results show that there was a clear difference between some spectra of the control and treatment groups, although in some cases, the spectra of the control group were closely related to the spectra of the treatment group.

Some spectral characteristics are rarely observed in raw FTIR spectra, but are observed after a preprocess based on the first or second derivative [17,18]. The second dendrogram (Figure 4B) analyzed the spectral regions containing all major bands (3629–2798 and 1701–1118 cm^−1^) in the first derivative of the spectrum with normalization and nine smoothing points (Figure 3A–C). The resulting dendrogram contains two main clusters. The cluster marked with red dashed lines contains 39 control group spectra and only 1 treatment group spectrum, shown as a red dashed line. On the other hand, two smaller clusters may be observed. One of them is characterized by a red dashed line that contains almost all spectra from the treatment group, and the second one contains the spectra of four patients (red dashed line), and 25 spectra from the control group are characterized by a blue dashed line. These two clusters (on the left) are therefore only slightly heterogeneous in their separation. Despite the heterogeneity, it is remarkable that the treatment group is isolated in a single cluster (on the right). The control group is highly differentiated, but there is good similarity in the treatment group.

The third dendrogram (Figure 4C) analyzed the spectra after preprocessing based on the second derivative with normalization and 17 smoothing points (Figure 3D–F). It included the whole spectral range of 4000–400 cm^−1^. This dendrogram confirms the conclusions of the previous HCA and shows even better separation of the spectra from the control and treatment groups. The cluster marked with red dashed lines contains only the spectra of the control group (26 spectra), while in the cluster marked with blue dashed lines, all the spectra of the treatment group and only a few spectra of the control group are classified. Thus, analysis of the spectra after transformation to the second derivatives, even considering the entire investigated spectral range, produced the best result for distinguishing between groups.

The other chemometric method applied in the study was PCA. It is an algorithm used to reduce the number of variables that describe a dataset. The obtained PCA results are presented in Figure 5. In PCA, points that represent a similar feature are located close together on a PCA score plot. PCA score plots are popular for visualizing spectral data sets obtained from this analysis.

Figure 5A shows the PCA results of n the entire spectral range for the second derivatives of the spectra with 17 smoothing points. It can be seen that in the PCA score plot, there is a partial separation of the groups along the first principal component (PC1). The groups overlap, but the spectra of the treatment group tend to cluster on the left and those of the control group on the right. Looking at the loading plots (Figure 6A), it can be noticed that the bands at approximately 1400 to 1700 cm^−1^ and individual bands at approximately 2800 to 3000 cm^−1^ had a large impact on the observed separation.

The PCA was then repeated, considering only the spectral ranges where potential changes were observed. The same preprocessing steps were applied, but the frequency ranges were 3700–2600 and 1800–900 cm^−1^ (the fingerprint region). The PCA score plot obtained after the analysis is shown in Figure 5B. It can be seen that there is a slight division of the groups along the second principal component (PC2). The spectra of the treatment group have lower PC2 values, whereas the spectra of the control group are higher. Moreover, it can be seen that the data obtained for the treatment group are more concentrated, indicating that this group contains spectra of greater similarity. However, the dispersion of the control group spectra is greater, which confirms the previously indicated large diversity of the control group. To evaluate which spectral bands affect the differences observed in the PCA score plot, a loading plot was prepared (Figure 6B). When comparing this loading plot with the average spectra (bottom plot in Figure 6B), it can be seen that there is a distinct change in the 3100–2900 cm^−1^ region. Overall, PCA showed worse results in terms of separating groups than HCA, but also confirmed that there were differences between the control and treatment groups and that the control group was significantly more heterogeneous.

## 3. Materials and Methods

In the present study, 40 hair samples of patients (26 women and 14 men) diagnosed with mood disorders such as depression or bipolar disorder were provided by the Department of Adult Psychiatry, in accordance with Bioethical Commission approval no. 1072.6120.302.2018. The healthy controls group included 75 volunteers (61 women and 14 men) without bipolar disorder or depression. All information, including clinical and demographic data for the control group (Appendix A) and the treatment group (Appendix A), are provided in the Appendix A. Data such as age, sex, type of disorder, used drugs, the dose, and hair colors were obtained. In the case of the control group, we only used age, sex, and hair color data. In addition, each person gave their informed written consent to the study. Each sample was taken from the occipital part of the head and stored in sealed polyethylene bags, at room temperature, in a dry and dark place until analysis. 

The reagents used throughout the experiments were: methanol purchased from Fluka Analytical (Seelze, Germany) and ultrapure water (18.2 MΩ cm, <3 ppb TOC) generated with the Milli-Q system from Merck-Millipore (Darmstadt, Germany), which was used to wash hair.

Samples taken from volunteers (reference database) and patients with clinically diagnosed depression were properly prepared before recording the ATR-FTIR spectra. Hair was washed based on the procedure used and described by Kim et al. [19]. This procedure was modified: It consisted of shaking the hair samples for two min successively in methanol, water, and again in methanol, with each step repeated twice. After washing, the samples were dried in a fume hood and centrifuged (2 min, 4000 rpm).

Sample analysis was performed on a Thermo Scientific FTIR Spectrometer (Carlsbad, CA, USA) with an ATR diamond ZnSe crystal. All spectra were collected using the 32-scans condition with a resolution of 4 cm^−1^ in the spectral range of 4000–650 cm^−1^ at 25 °C. The background was taken without placing the sample on the crystal. The pretreatment methods, including baseline correction, normalization (vector normalization), and derivative (first- and second-order), were performed on all spectra before applying chemometric methods. Data analysis was performed with the use of OPUS software and RStudio (Boston, MA, USA) (version 3.6.1) with the ChemoSpec 6.1.0 package.

## 4. Conclusions

In conclusion, this preliminary study provides insight into hair analysis using ATR-FTIR spectroscopy to find differences between a control group and a group of patients with mood disorders. In the case of changes in the hair structure, no changes were found in the bands derived from the hair-building proteins. Hair is primarily made of proteins, but also contains other ingredients such as melanin, lipids, and minerals. Some changes in the bands derived from CH_2_/CH_3_ were observed, including a significant decrease in the bands’ intensity in the range of approx. 2800–3100 cm^−1^ and smaller differences at around 1460 cm^−1^. This was likely due to a decrease in the amount of lipids in the treatment group or slight structural changes in proteins (amino acid side chains). 

Therefore, we concluded that, on the basis of the average spectra, the control group could be initially distinguished from the group of patients with mood disorders. However, the standard deviation for the spectra from the control group was large, so the diversity of this group was high.

Further chemometric analysis using HCA and PCA methods led to similar conclusions. The control group was very diverse, and its data partially overlapped with the data of the treatment group. On this basis, it is difficult to conclude that mood disorders are a factor that can unequivocally affect hair structure. Hair is likely influenced by many factors, such as age, genetics, hygiene, and the amount of stress in daily life. However, both chemometric methods used showed that there were some differences between the control and treatment groups; in particular, the HCA analysis performed for the second derivatives of the spectra showed the best separation of the groups.

To summarize, the preliminary study provides many interesting conclusions. Establishing the exact reasons for the above may help in the preparation of a fast and easy pre-diagnosis test for mood disorder, emphasizing that examination of the hair structure with the ATR-FTIR method is noninvasive and relatively inexpensive.

## Figures and Tables

**Figure 1 molecules-27-05318-f001:**
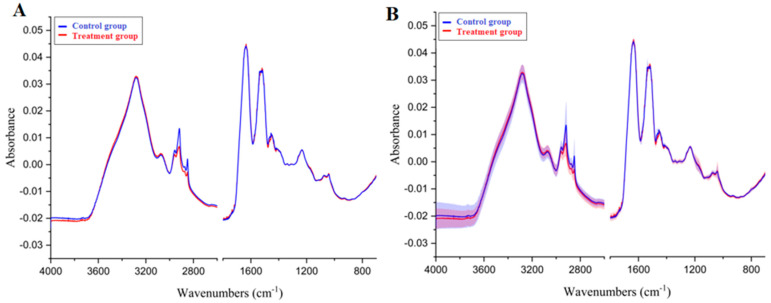
Comparison of the hair FTIR spectra based on the mean without standard deviation (**A**) or the mean with standard deviation marked (**B**). The mean of the control group spectra is colored blue; the mean of the treatment group spectra is colored red.

**Figure 2 molecules-27-05318-f002:**
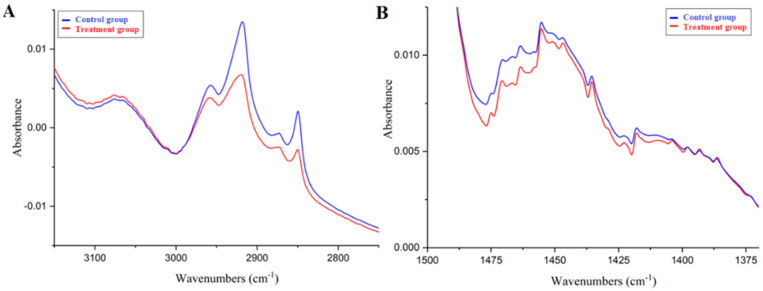
Comparison of hair FTIR spectra: the mean of the high (**A**) and low (**B**) range of the spectrum in which the changes were observed. The mean of the control group spectra is colored blue; the mean of the treatment group spectra is colored red.

**Figure 3 molecules-27-05318-f003:**
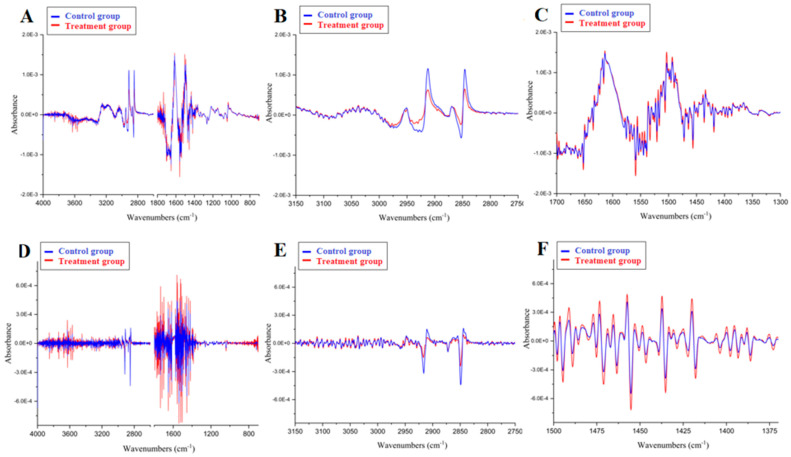
The first (**A**–**C**) and second (**D**–**F**) derivatives of mean spectra are presented in Figure 1 and Figure 2. The mean of the control group spectra is colored blue; the mean of the treatment group spectra is colored red.

**Figure 4 molecules-27-05318-f004:**
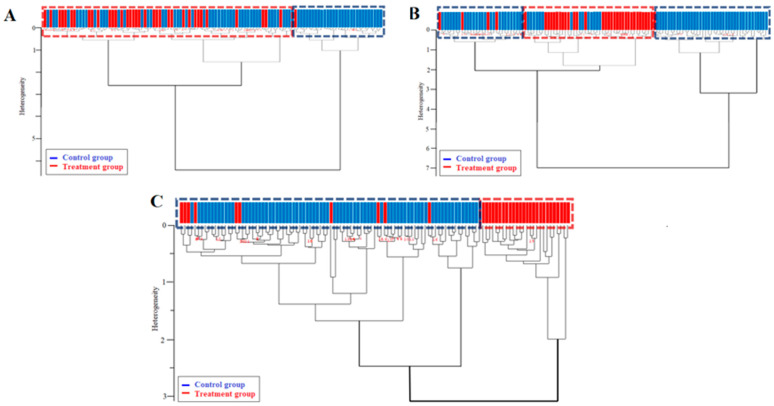
Dendrograms obtained after hierarchical cluster analysis of hair spectra, in the frequency range 3100–2800 cm^−1^ and 1470–1400 cm^−1^, output spectrum with vector normalization (**A**); in the frequency range 3629–2798 cm^−1^ and 1707–1118 cm^−1^, first derivative with vector normalization and 9 smoothing points (**B**); and in the frequency range 4000–400 cm^−1^, second derivative with vector normalization and 17 smoothing points (**C**). The red color indicates the patient group, whereas the blue color indicates the control group.

**Figure 5 molecules-27-05318-f005:**
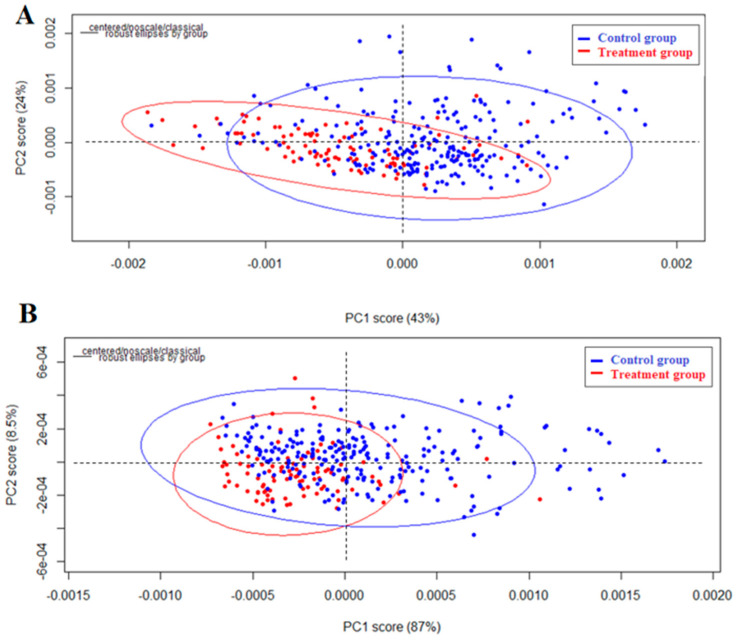
PCA score plots of hair spectra: in the frequency range of 4000–400 cm^−1^, second derivative with 17 smoothing points (**A**); and in the frequency ranges of 3700–2600 and 1800–900 cm^−1^, second derivative with 17 smoothing points (**B**). The red color indicates the treatment group, whereas the blue color indicates the control group.

**Figure 6 molecules-27-05318-f006:**
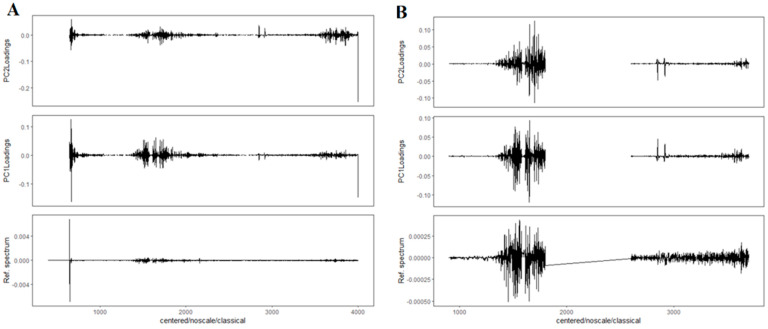
Loading plots corresponding to the appropriate PCA score plots are shown in Figure 5A (**A**) and Figure 5B (**B**).

**Table 1 molecules-27-05318-t001:** Assignment of hair FTIR spectral bands based on the literature [12].

Band	Frequency (cm^−1^)	Description
Amide A	3300	NH stretching
Amide B	3100	NH stretching
Amide I	1690–1600	C=O stretching
Amide II	1575–1480	CN stretching; NH bending
Amide III	1301–1229	CN stretching; NH bending
Amide IV	767–625	OCN bending
Amide V	800–640	Out-of-plane NH bending
Amide VI	606–537	Out-of-plane C=O bending
Amide VII	200	Skeletal torsion
C–H stretching	3060–2850	CH_2_ and CH_3_ antisymmetric and symmetric stretching vibration modes
C–H deformation	1470–1400	CH_2_ and CH_3_ bending modes
O–C stretching	1106–991	Ester functional group
C–O stretching	3430–3090	Alcohol functional group
Stretching –SO	1071	Cysteine oxide
Stretching –SO_2_	1121	Cysteine dioxide
Stretching –SO_3_	1040	Cysteic acid

## Data Availability

Not applicable.

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
