# Peer review of "Fast and Noninvasive Hair Test for Preliminary Diagnosis of Mood Disorders"

_molecules, 2022, doi:10.3390/molecules27165318_

Round 1
Reviewer 1 Report
In the manuscript” Fast and non-invasive hair test for preliminary Diagnosis of Mood Disorders”, the authors developed a potential test for fast and non-invasive analysis of hair samples to initially classified healthy and with mood disorders subjects based on their hair sample. That’s very interesting. However, the following issues are need to be addressed and clarified:
Introduction:
1. Please provide references for Page 2, line 62-66.
2. Since it has been mentioned repeatedly in the background that women are more likely to suffer from depression than men, why was it not shown in the subsequent analysis?
3. Page 2, line 80, please determine whether the method is micro-invasive or non-invasive.
Results and discussion:
4. References do not seem to correspond to the manuscript, e.g. 14 and 17. Please refer to the normative literature.
5. Please provide references for Page 3, line 104-108. And I disagree the age and gender of the sample donor do not influence the hair spectrum, and the literature the authors refer to does not prove the point.
6. Why were nearly 10% of the spectra discarded? I think there is something wrong with the method the author processed the samples. It seems that the detection position cannot be located by using ATR directly to detect the hair. Therefore, it is suggested to use microscopic IR.
7. Please add legend to the figures.
8. The identification and interpretation of the band is not detailed enough, especially the derivative preprocessing part. There are too many frequency ranges, so it is recommended to select only 3700-2600 and 1800-900 cm-1 for analysis.
9. Why only unsupervised algorithms are used? Please use supervised machine learning algorithms for modeling.
Materials and methods:
10. As mentioned above, I don't think it's scientific to test hair directly with ATR, why not try microscopic IR or KBr pellet?
Supplementary Materials:
11. The table number does not match the manuscript, and in table S1, ‘K’ should be changed to ‘W’?
Author Response
Reviewer #1:
In the manuscript” Fast and non-invasive hair test for preliminary Diagnosis of Mood Disorders”, the authors developed a potential test for fast and non-invasive analysis of hair samples to initially classified healthy and with mood disorders subjects based on their hair sample. That’s very interesting. However, the following issues are need to be addressed and clarified:
Introduction:
- Please provide references for Page 2, line 62-66.
The Authors have added references in accordance with the suggestion of the Reviewer.
- Since it has been mentioned repeatedly in the background that women are more likely to suffer from depression than men, why was it not shown in the subsequent analysis?
It is a common fact that more women suffer from depression than men. The authors mention it because it can be noticed in the studied group of patients, 26 women, 14 men. The proposed studies did not focus on gender division, as these are preliminary studies. Indeed, it is worth conducting a discriminatory analysis taking into account sex, but for this, the same number of samples from women and also men are required.
- Page 2, line 80, please determine whether the method is micro-invasive or non-invasive.
The Authors agree with the Reviewer, of course, it should be non-invasive. It has been changed in the manuscript.
Results and discussion:
- References do not seem to correspond to the manuscript, e.g. 14 and 17. Please refer to the normative literature.
The Authors are very grateful to the Reviewer for this valuable attention. There was indeed an error in the order of the cited references. It has been verified and corrected in the manuscript.
- Please provide references for Page 3, line 104-108. And I disagree the age and gender of the sample donor do not influence the hair spectrum, and the literature the authors refer to does not prove the point.
In the quoted publication, Pienpinijtham et al. [15] write, “However, there is no correlation between age/gender and spectral feature." On this basis, the Authors claim that the age and gender of the sample do not influence the hair spectrum. A correct reference has been added to the manuscript.
[14] Pienpinijtham, P., Thammacharoen, C., Naranitad, S., Ekgasit, S. Analysis of cosmetic residues on a single human hair by ATR FT-IR microspectroscopy. Spectrochimica Acta Part A: Molecular and Biomolecular Spectroscopy, 197, (2018), 230–236.
- Why were nearly 10% of the spectra discarded? I think there is something wrong with the method the author processed the samples. It seems that the detection position cannot be located by using ATR directly to detect the hair. Therefore, it is suggested to use microscopic IR.
The 10% of the spectra were discarded because, as the Reviewer suggested, the light beam is not always hitting the hairs correctly and the spectra do not resemble hair spectra. Therefore, we provided three repeats of each sample (n=3) to eliminate this problem. Even with a perfect method, there are some discarded samples/spectra. The use of the ATR-FTIR technique was dictated by the desire to develop a simple, fast, and minimal sample preparation method. The application of the IR microscopy technique would require a prior cutting and appropriate sample preparation; therefore, a simpler technique was chosen.
- Please add legend to the figures.
The legend has been added to all figures.
- The identification and interpretation of the band are not detailed enough, especially the derivative preprocessing part. There are too many frequency ranges, so it is recommended to select only 3700-2600 and 1800-900 cm-1 for analysis.
The Authors thank the Reviewer for these suggestions. Preprocessing of the spectra was done on the whole spectra range (4000-400 cm-1), and the Authors did it only for protein. Concerning the above remark, the appropriate information was added to Table 1. in the manuscript.
- Why only unsupervised algorithms are used? Please use supervised machine learning algorithms for modeling.
The Authors are grateful for this proposition. It could be interesting in the future to develop an automated method for identifying the changes described in this article, or even classifying patients with a mood disorder to support the diagnosis. However, since this was a preliminary study and the number of positive/negative examples (i.e. spectra) was not enough to train and test any supervised machine learning algorithm (e.g. SVM, or ANN). Thus, we decided to use only unsupervised algorithm methods.
Materials and methods:
- As mentioned above, I don't think it's scientific to test hair directly with ATR, why not try microscopic IR or KBr pellet?
The Authors are grateful for the Reviewer's advice. It is a really good idea to try microscopic IR for future research because it is possible to take screening of different areas of the hair. ATR compared to KBr pellets is a relatively simpler and faster experiment, giving similar results. However, as mentioned above, our study aimed to develop a method that requires minimal sample preparation.
Suplementarny Materials:
- The table number does not match the manuscript, and in table S1, ‘K’ should be changed to ‘W’?
The Authors strongly agree with the Reviewer's suggestion. It has been changed in the supplementary material.

Reviewer 2 Report
The authors presented a work designed to disclose potential biomarkers for preliminary diagnosis of mood disorders. In my opinion, the aim is very interesting and relevant, however I have some concerns regarding the methodology used, as it is not very informative on the alterations observed and, as such, these results are utterly preliminary. Other concerns relate to selection of subjects to be included in the groups (controls vs. patients); as no statistical analysis was provided to support homogeneity of the groups (sex, age, dietary habits, etc) this might substantiate difficulties in the analysis and interpretation of data. Of note, it is uncertain if the alterations reported derive from the disease itself or from the administration of antidepressants in the patients’ group (no drugs were reported for controls and it is expected they do not use these type of pharmaceuticals). No information was provided on the sample collection.
Other specific comments:
Title: “Fast and non-invasive hair test for preliminary Diagnosis of Mood Disorders”. In title and throughout the document, the authors refer “mood disorders”. However, only depressed patients were studied.
Bibliography: References provided are not actual – only 25% of them are from the last 5 years! And many of the other are very old, including some related to prevalence of the diseases. Line 32-33: Provide a current relevant reference for prevalence, in this sentence (references 1 and 2 are from 2012 and 2013). Some references are not properly provided (e.g., REF 5 is a thesis but is not acknowledged as that).
I paid special attention to the Abstract section as this works as an invitation for full article reading:
* Lines 15-16: In the Reviewer’s opinion, the main aim of the study should be the development of a test for fast, non-invasive diagnosis of mood disorders (depression). The use of hair samples of both healthy individuals and individuals with mood disorders is not a aim – it is a mean (methods section) used for validation, but it is not the aim itself.
* Lines 17-18: The database included 75 control subjects (who were not diagnosed with depression#), and 40 patients diagnosed with mood disorders such as depression or bipolar disorder #do the object of study are patients with depression (only) or mood disorders? This should be clarified and consistently presented throughout the document.
* Lines 18-19: “Both women and men, aged 18–65 have participated in the research”. It should be referred instead that both groups were homogeneous/balanced (in terms of age, sex, other characteristics). Does the depression-diagnosed patient make any medication? Could this be relevant for the obtained results, as it is assumed that the healthy patients have no need of pharmaceuticals?
* Lines 18-19: After the hair samples were taken, they were [properly prepared and additionally] washed – What do the authors mean by “properly prepared”? I suggest specifying this or removing of the text inside brackets as it does not provide any relevant information.
* Lines 24-25: As a consequence of the conducted research, potential differences were noticed. There was a visible change in the spectra intensity at 25 around 2800–3100 cm-1, and smaller differences around 1460 cm-1. – This is highly vague. There are any molecules that could be assigned to this profile?
Line 30: Suggestion: correct “PCA analysis”
Many typos throughout the manuscript. Some examples: * Line 32: “mood disorders, are” – Suggestion: remove comma. * Line 35: life. [1,2]. – Suggestion: remove dot.
Line 32: I suggest moving the first sentence after the description of “mood disorders” as this refers to prevalence, the second one to disease definition, and the third one to prevalence again.
Line 46: Clarify “problematic” (perhaps, the authors mean complex)
Line 48: “sometimes lasting up to several days” – the diagnosis? But it is not made instantaneously, based on the interview and “patient’s clinical picture”? Please clarify
Line 49: As no other matrices were suggested, I believe “alternative” does not fit. Instead of alternative I suggest non-invasive. In line 50, I suggest removing “alternative”
Lines 59-60: “In the hair of people with mental disorders, endogenous substances such as cortisol, which is produced in the body in response to stress, can also be determined, and this may serve as a potential biomarker of depression [7,8].” These references do not seem to support cortisol as a biomarker for depression. In fact, assumptions only based in cortisol levels seem exaggerated as cortisol is not specific at all for this condition and might be elevated in several other circumstances. In my opinion, this concern might be at least acknowledged in the paper. In this sense, are there any other potential markers that, all combined, could provide a better picture of the patient’s condition?
Lines 62-64: “The studies carried out so far have shown that the multi-element hair profile of people with depression who tried to commit suicide (the most severe form of depression) is different compared to people who are apparently healthy”. Please clarify “different” and provide references for this statement.
Line 66: “which have been reported in patients with depression” Suggestion: “whose dysregulation has been reported”
Line 67: “The research also showed that women are much more likely (in a ratio of 2.5 : 1) to suffer from depression than men [9-11]”. In my opinion, this sentence should be moved to the beginning of the introduction – where prevalence is described, and be supported by references relating to prevalence (instead of REFs 9-11).
Line 73-74: “Research has already been carried out on the use of ATR-FTIR spectroscopy to study hair samples” This is repeating the previous sentence.
Line 75: “it has confirmed the concept of a change” this is vague. It is highly important to clarify the biological change. Which particular lipids changed? What is the reason for that increase? The cancer itself or any cancer-related medication?
Lines 76-77: “Patients with breast cancer showed an increase in the band maximum ratio at 1446–1456 cm-1 for C-H bending vibrations in the ATR-FTIR spectra of a single hair fiber [12]” And this is indicative of what?
Methods for statistical analysis should be provided in the “materials and methods section”
Why was this methodology selected for the analysis, as it does not provide evidence of specific molecular alterations?
Lines 241-242; “In the patient's group, was obtained data such as age, sex, used drugs, the dose, dosage, and hair colours. In the case of the control group only age, sex, and hair color”. Regarding these characteristics, are the samples (control vs. patients’ group) homogeneous? Statistical analysis confirming these issues should be provided. Even so, the lack of information on used drugs/pharmaceuticals, dietary habits, etc. might represent a significant limitation to the study as these could impair data interpretation.
Line 244: Suggest changing to “Each sample was”. “Each samples were specially prepared and stored in sealed polyethylene bags”. What do the authors mean by “specially prepared”? The authors provide no information on collection of the samples (e.g., hair segment length cut/used).
Line 255: “centrifuged (2 min, 4000 rpm)” What was the aim of this centrifugation? Was the whole segment of the hair used for analysis?
Author Response
Reviewer #2:
The authors presented a work designed to disclose potential biomarkers for preliminary diagnosis of mood disorders. In my opinion, the aim is very interesting and relevant, however I have some concerns regarding the methodology used, as it is not very informative on the alterations observed and, as such, these results are utterly preliminary. Other concerns relate to selection of subjects to be included in the groups (controls vs. patients); as no statistical analysis was provided to support homogeneity of the groups (sex, age, dietary habits, etc) this might substantiate difficulties in the analysis and interpretation of data. Of note, it is uncertain if the alterations reported derive from the disease itself or from the administration of antidepressants in the patients’ group (no drugs were reported for controls and it is expected they do not use these type of pharmaceuticals). No information was provided on the sample collection.
The Authors are grateful for the review. Patients and people from the control group were included in the study that met the appropriate criteria, for example, healthy people declared that they did not use and had never been treated for mood disorders. While patients provided information on drugs and doses that they use during pharmacotherapy. Therefore, the patterns of active substances of these drugs were also tested and compared with the obtained hair spectra. The reported differences are not drug-related. Information about sample collection has been added to the manuscript. A detailed explanation of all questions is provided below.
Other specific comments:
- Title: “Fast and non-invasive hair test for preliminary Diagnosis of Mood Disorders”. In title and throughout the document, the authors refer “mood disorders”. However, only depressed patients were studied.
The Authors are grateful for the point that the Reviewer has raised. In the presented research, the Authors studied a group of patients with known mood disorders, including both patients with bipolar disorder (characterized by an alternating episode of depression and mania) and with a unipolar disorder called depression. Thus, in both conditions, there is a depressive episode. Therefore, it may seem that changes were only observed in depression. To clarify this, we need more patient samples, which we are working on constantly. Information about the patient's type of disorder is provided in Table S2. in the supplementary materials.
- Bibliography: References provided are not actual – only 25% of them are from the last 5 years! And many of the other are very old, including some related to prevalence of the diseases. Line 32-33: Provide a current relevant reference for prevalence, in this sentence (references 1 and 2 are from 2012 and 2013). Some references are not properly provided (e.g., REF 5 is a thesis but is not acknowledged as that).
The Authors thank the Reviewer for this question. Some references have been changed in the manuscript.
I paid special attention to the Abstract section as this works as an invitation for full article reading:
- Lines 15-16: In the Reviewer’s opinion, the main aim of the study should be the development of a test for fast, non-invasive diagnosis of mood disorders (depression). The use of hair samples of both healthy individuals and individuals with mood disorders is not a aim – it is a mean (methods section) used for validation, but it is not the aim itself.
The Authors strongly agree with the Reviewer's suggestion. The main objective of the study was to develop a potential test for fast and non-invasive diagnosis of mood disorders based on the analysis of hair samples. It has been changed in the manuscript.
- Lines 17-18: The database included 75 control subjects (who were not diagnosed with depression#), and 40 patients diagnosed with mood disorders such as depression or bipolar disorder #do the object of study are patients with depression (only) or mood disorders? This should be clarified and consistently presented throughout the document.
The Authors are grateful for the Reviewer's suggestion. In the presented research, hair samples were used from a group of patients with mood disorders with both bipolar disorder and unipolar disorder called depression. Information about the type of disorder of the patient is provided in Table 2 of the supplementary materials.
- Lines 18-19: “Both women and men, aged 18–65 have participated in the research”. It should be referred instead that both groups were homogeneous/balanced (in terms of age, sex, other characteristics). Does the depression-diagnosed patient make any medication? Could this be relevant for the obtained results, as it is assumed that the healthy patients have no need of pharmaceuticals?
Patients and people from the control group were included in the study that met the appropriate criteria, for example, healthy people declared that they did not use and had never been treated for mood disorders. Furthermore, the patients were recruited by a specialist psychiatrist. Table S2 shows the type and dose of drugs taken in therapy. Therefore, the drug patterns were also measured and compared with the patient spectra, and this is not relevant to the results obtained. The observed changes are not the result of the drugs used.
- Lines 18-19: After the hair samples were taken, they were [properly prepared and additionally] washed – What do the authors mean by “properly prepared”? I suggest specifying this or removing of the text inside brackets as it does not provide any relevant information.
The Authors are grateful to the Reviewer for this advice. It has been changed in the manuscript.
- Lines 24-25: As a consequence of the conducted research, potential differences were noticed. There was a visible change in the spectra intensity at 25 around 2800–3100 cm-1, and smaller differences around 1460 cm-1. – This is highly vague. There are any molecules that could be assigned to this profile?
The Authors are grateful for the valuable Reviewer's suggestion. There are bands that can be assigned to specific protein vibrations and are enumerated in Table 1. It is important, so to clarify, it has been changed in the article.
- Line 30: Suggestion: correct “PCA analysis”
The Authors thank the Reviewer for noticing this. It has been changed in the manuscript.
Many typos throughout the manuscript. Some examples:
- Line 32: “mood disorders, are” – Suggestion: remove comma.
- Line 35: life. [1,2]. – Suggestion: remove dot.
- Line 32: I suggest moving the first sentence after the description of “mood disorders” as this refers to prevalence, the second one to disease definition, and the third one to prevalence again.
- Line 46: Clarify “problematic” (perhaps, the authors mean complex)
- Line 49: As no other matrices were suggested, I believe “alternative” does not fit. Instead of alternative I suggest non-invasive. In line 50, I suggest removing “alternative”
- Line 66: “which have been reported in patients with depression” Suggestion: “whose dysregulation has been reported”
- Line 67: “The research also showed that women are much more likely (in a ratio of
5 :1) to suffer from depression than men [9-11]”. In my opinion, this sentence should be moved to the beginning of the introduction – where prevalence is described, and be supported by references relating to prevalence (instead of REFs 9-11). - Line 73-74: “Research has already been carried out on the use of ATR-FTIR spectroscopy to study hair samples” This is repeating the previous sentence.
- Line 75: “it has confirmed the concept of a change” this is vague. It is highly important to clarify the biological change. Which particular lipids changed? What is the reason for that increase? The cancer itself or any cancer-related medication?
- Lines 76-77: “Patients with breast cancer showed an increase in the band maximum ratio at 1446–1456 cm-1 for C-H bending vibrations in the ATR-FTIR spectra of a single hair fiber [12]” And this is indicative of what? Methods for statistical analysis should be provided in the “materials and methods section” Why was this methodology selected for the analysis, as it does not provide evidence of specific molecular alterations?
- Line 244: Suggest changing to “Each sample was”. “Each samples were specially prepared and stored in sealed polyethylene bags”. What do the authors mean by “specially prepared”? The authors provide no information on collection of the samples (e.g., hair segment length cut/used).
The Authors strongly agree with the Reviewer’s advice. Each suggestion has been added to the manuscript.
- Line 48: “sometimes lasting up to several days” – the diagnosis? But it is not made instantaneously, based on the interview and “patient’s clinical picture”? Please clarify
The Authors are agree with the Reviewer, the initial diagnosis is made right away, on the basis of the interview and the "clinical picture of the patient." However, in some cases, when there are difficulties in diagnosing the disease, additional tests should be performed, such as blood counts or the determination of disease biomarkers.
- Lines 59-60: “In the hair of people with mental disorders, endogenous substances such as cortisol, which is produced in the body in response to stress, can also be determined, and this may serve as a potential biomarker of depression [7,8].” These references do not seem to support cortisol as a biomarker for depression. In fact, assumptions only based in cortisol levels seem exaggerated as cortisol is not specific at all for this condition and might be elevated in several other circumstances. In my opinion, this concern might be at least acknowledged in the paper. In this sense, are there any other potential markers that, all combined, could provide a better picture of the patient’s condition?
The Authors are grateful to the Reviewer for this advice. References have been verified and corrected in the manuscript.
- Lines 62-64: “The studies carried out so far have shown that the multi-element hair profile of people with depression who tried to commit is different compared to people who are apparently healthy”. Please clarify “different” and provide references for this statement.
The Authors agree with the Reviewer's advice, and it should be clarified. The studies described by Momčilović et al. [8,9] show that the multi-bioelement profile in the hair of the depressed subjects who attempted suicide (the heaviest form of depression) is different when compared to the apparently healthy subjects.
[8] B. Momčilović, J. Prejac, A.V. Skalny, N. Mimica, in search of decoding the syntax of the bioelements in human hair – A critical overview, J. Trace Elem. Med. Biol. 50 (2018) 543–553.
[9] B. Momčilović, J. Morović, J. Prejac, A.V. Skalny, N. Ivičić, Trace element profile of human depression – the tapestry of patterns, Trace Elem. Electrolytes 25 (2008) 187–190
- Lines 241-242; “In the patient's group, was obtained data such as age, sex, used drugs, the dose, dosage, and hair colours. In the case of the control group only age, sex, and hair color”. Regarding these characteristics, are the samples (control vs. patients’ group) homogeneous? Statistical analysis confirming these issues should be provided. Even so, the lack of information on used drugs/pharmaceuticals, dietary habits, etc. might represent a significant limitation to the study as these could impair data interpretation.
Participants in the control group were declared to have not used a drug used in the treatment of mood disorders and had never been treated for mood disorders. Therefore, the Authors assume the homogeneity of the samples. In the case of dietary habits, the Authors have no information. But it is a really important suggestion for the future because this research will be modified and repeated on a larger group of people. Then, you should take information about, for example, eating habits or supplementation.
- Line 255: “centrifuged (2 min, 4000 rpm)” What was the aim of this centrifugation? Was the whole segment of the hair used for analysis?
The Authors are grateful for the valuable Reviewer's suggestion. In research the Authors used the procedure described by C.H. Kim J. et al [12]. Of course, it was modified. In the original citation, the hairs were washed with methanol and distilled water. The Authors used centrifugation in order to obtain a slightly better effectiveness of the cleaning/washing process from any residue on the hairs. In that way, it is more probable that the results from the ATR measurement are not affected by impurities.
[12] C.H. Kim J., Lee S., In S., Choi H., Validation of a simultaneous analytical method for the detection of 27 benzodiazepines and metabolites and zolpidem in hair using LC–MS/MS and its application to human and rat hair, J. Chromatogr. 879 (2011) 878-86.
Round 2
Reviewer 1 Report
no comments.